# Effects of Capsaicin on Glucose Uptake and Consumption in Hepatocytes

**DOI:** 10.3390/molecules28135258

**Published:** 2023-07-06

**Authors:** Haolong Zeng, Nian Shi, Wenlei Peng, Qing Yang, Jingnan Ren, Hong Yang, Lingling Chen, Yijie Chen, Jun Guo

**Affiliations:** 1Department of Laboratory Medicine, Tongji Hospital, Tongji Medical College, Huazhong University of Science and Technology, Wuhan 430030, China; zenghaolong@tjh.tjmu.edu.cn; 2College of Food Science and Technology, Huazhong Agricultural University, Wuhan 430070, China; shinian0916@163.com (N.S.); renjingnan@mail.hzau.edu.cn (J.R.); 3Hubei Key Laboratory of Agricultural Bioinformatics, College of Informatics, Huazhong Agricultural University, Wuhan 430070, China; 15926352256@163.com (W.P.); llchen@mail.hzau.edu.cn (L.C.); 4Key Laboratory for Deep Processing of Major Grain and Oil of Ministry of Education, Wuhan Polytechnic University, Wuhan 430023, China; qingyang@whu.edu.cn; 5Department of Emergency, Wuhan Municipal Fourth Hospital Affiliated Puai Hospital, Tongji Medical College, Huazhong University of Science and Technology, Wuhan 430034, China; jerrymr@126.com; 6Jiangsu Provincial Key Laboratory of Critical Care Medicine, Department of Critical Care Medicine, Zhongda Hospital, School of Medicine, Southeast University, Nanjing 210009, China; 7Department of Critical Care Medicine, Union Jiangbei Hospital, Huazhong University of Science and Technology, Wuhan 430100, China

**Keywords:** capsaicin, HepG2, glucose, APT, transcriptomes

## Abstract

Obesity represents a major health challenge because it substantially increases the risk of metabolic diseases. Capsaicin, the major active ingredient of *Capsicum* spp., has been reported to possess anti-obesity activity. Hereon, the effect of capsaicin on glucose uptake and consumption in hepatocytes was extensively studied. Capsaicin was shown to accelerate the glucose uptake/consumption and the ATP production of hepatocytes. The elevation of intracellular Ca^2+^ was thought to be a potential mechanism. By transcriptome analysis, 78, 146 and 507 differentially expressed genes (DEGs) were identified between capsaicin and the control group for 4 h, 12 h and 24 h treatments. Kyoto Encyclopedia of Genes and Genomes (KEGG) analysis showed that most of the DEGs were involved in canonical pathways, like MAPK and PI3K-AKT signaling pathways. Clustering analysis showed that many DEGs were associated with glucose and amino acid metabolism. The variation trend in genes related to glucose and amino acid metabolism (like CTH, VEGFA, PCK2 and IGFBP3) in the quantitative PCR (q-PCR) assay was consistent with the transcriptome data. These results demonstrated that capsaicin efficiently accelerated the glucose uptake and consumption of hepatocytes.

## 1. Introduction

Obesity is a state of metabolic disorder, which is a consequence of increased energy intake or decreased energy expenditure. Nowadays, it is a major public health concern around the world [1]. This metabolic disorder does not only impair people’s quality of life, but also causes many serious health problems like non-alcoholic fatty liver, type 2 diabetes, cardiovascular disease, hyperlipemia and cancer [2]. Therefore, prevention and treatment of obesity is a common health challenge worldwide. At present, medications are widely used to combat obesity but with limited effect, with even undesirable side effects [3]. Thus, intervention of natural bioactive components on body health, especially obesity, has received much attention due to their multiple health benefits with relatively low toxicity.

Capsaicin (trans-8-methyl-N-vanillyl-6-nonenamide, CAP) is the major pungent compound from the fruit of *Capsicum* spp. (hot pepper), which is responsible for the hot and spicy flavor. It is also the major bioactive compound in chili peppers for possessing anti-oxidative, anti-inflammatory, anti-cancer, cardio-protection and anti-obesity activities [4,5,6]. Among these functions, the direct influence of capsaicin on lipid and energy metabolism has been widely studied in recent years. CAP has been demonstrated to inhibit adipogenesis and increase oxidation in 3T3-L1 via regulating the expression of related genes [7]. In Caco-2 cells, capsaicin could increase the expression of glycolytic enzymes, which are related to the energy metabolism of cells [8]. Dietary capsaicin is also reported to ameliorate diet-induced obesity in rodents and humans by enhancing energy metabolism, supporting the role as an anti-obesity compound. The inappropriate dosage of CAP ingestion has adverse health effects on human subjects, like gastric ulcers, hyperalgesia, and gut inflammation, and even promotes the development of tumors [9,10,11,12].

The transient receptor potential channel vanilloid type 1 (TRPV1), which was expressed in many tissues, is a member of the transient receptor potential (TRP) channel superfamily. The activation of TRP has been shown to be involved in a wide range of metabolic processes, including glucose metabolism and the progression of obesity [13]. The liver plays an essential role in nutrient metabolism as it has first access to the ingested nutrients by virtue of their absorption into the hepatic portal vein. It is the main metabolic organ, especially in glycometabolism. The dysregulation of hepatic glucose metabolic pathways is central to many pathologies, including obesity and even type 2 diabetes [14]. It has demonstrated that TRPV1 is a mechanosensitive ion channel that has been shown to regulate the energy expenditure and cholesterol metabolism in hepatocytes [15,16]. As a highly selective agonist for TRPV1, CAP exerts many of its biological activities through TRPV1 activation [17]. It was also reported to exert action on the TRPV1 channel to mediate the regulation of glucose homeostasis and obesity.

However, few investigations have been carried out on the influence of capsaicin on glucose metabolism in human hepatic cells. The underlying mechanisms by which capsaicin affects the glucose homeostasis of liver have not been completely clarified yet. Thus, in this research, we studied the effect of CAP on glucose uptake and consumption in HepG2 cells. Through RNA-seq combined q-PCR verification, the underline mechanism was also elucidated.

## 2. Results

### 2.1. The Impact of Capsaicin on Glucose Uptake and the Consumption of HepG2 Cells

In this study, we confirmed that capsaicin did not show any cytotoxicity against HepG 2 cells at 25~200 μM (Figure 1). Under the same condition, we measured glucose uptake in HepG2 cells. 2-NBDG is a fluorescent indicator for direct glucose uptake measurements [18]. In this research, we used 2-NBDG to study the impact of capsaicin on the glucose uptake of HepG 2 cells. The results showed that, generally, the 2-NBDG fluorescence intensity was significantly increased in capsaicin-treated HepG2 cells in a concentration-dependent manner, with the highest rate of 138.2% observed at 100 μM (Figure 2A). Glucose consumption is a significant indicator of energy metabolism and was monitored to assess the input of glycosis. Glucose concentration changes in the culture medium for capsaicin treatment were detected by a glucose kit to reflect whether capsaicin stimulation affected the glucose consumption of HepG2 cells. As shown in Figure 2B, at the concentration range of 25~200 μM, with the increase in capsaicin concentration, the glucose consumption of HepG2 cells also increased in a concentration-dependent manner. Compared with the DMSO control, the glucose consumption of 100 μM-treated cells reached 5.79 mM. When the capsaicin reached up to 200 μM, the glucose consumption increased to 6.08 mM, showing no difference between 100 mM capsaicin-treated cells. Therefore, 100 μM of capsaicin was selected for the following experiments. In addition to capsaicin concentration, the treatment time also had a significant effect on the glucose consumption of HepG2 cells. At the 100 μM treatment, with the prolongation of processing time, the glucose consumption of HepG2 cells gradually increased in a time-dependent manner (Figure 2C). At different time points, the glucose consumption of capsaicin-treated cells was higher than the DMSO control. At 12 h and 24 h treatments, the difference in glucose consumption compared with the DMSO-treated group was significant (Figure 2C). These results reflect that capsaicin can promote glucose consumption in hepatocytes. 

### 2.2. Effect of Capsaicin on the Intracellular ATP Production of HepG2 Cells

The results of glucose consumption indicated that capsaicin could accelerate the energy metabolism of HepG2 cells. To further investigate the effects of capsaicin on energy production, the intercellular ATP content was evaluated using an ATP assay kit. As shown in Figure 3A, compared with the DMSO control, capsaicin remarkably increased the ATP production of HepG2 cells in a dose-dependent manner, with the highest rate of 456% observed at 200 μM. The ATP productions of HepG2 cells treated with 100 μM at four different times (0, 4, 12 and 24 h) were detected to determine whether capsaicin has a time-dependent effect on ATP production in human hepatic cells. With the prolongation of processing times, the ATP production of HepG2 cells gradually increased in a time-dependent manner (Figure 3B). The ATP production of hepatocytes was not significantly affected by the 100 μM capsaicin treatment for 4 h and 8 h. As the processing time increased to 12 h, the level of ATP production in capsaicin was about 28% higher than that in cells treated with DMSO; this difference was significant (*p* < 0.05) (Figure 3B). As the time continued to extend to 24 h, ATP increased by 77% and showed a highly significant difference compared with the control (Figure 3B). These results showed the same trend as the effect of capsaicin treatment on the glucose consumption of hepatocytes.

### 2.3. The Involvement of Ca^2+^ in the Capsaicin-Induced Glucose Metabolism Increase in HepG2 Cells

Previous studies have reported that after inducing the activation of TRPV1 channel, capsaicin increased the intracellular calcium levels in Caco-2 cells and 3T3-L1 cells [19,20]. In our research, the effects of capsaicin on intracellular calcium levels of human hepatocytes were detected using Ca^2+^ sensitive dye Fluo3-AM. As shown in Figure 4A, capsaicin could promote the intracellular Ca^2+^ levels. Cells treated with 100 μM capsaicin emitted the strongest florescence, while the fluorescence signals emitted of lower concentration capsaicin-treated cells were relatively weaker.

BAPTA-AM is a calcium-chelating agent. BAPTA-AM was used to block the intracellular Ca^2+^ levels to elucidate whether an capsaicin-induced intracellular Ca^2+^ increase was involved in glucose consumption. As shown in Figure 4B, compared with cells treated with capsaicin alone, BAPTA-AM significantly inhibited the glucose consumption of HepG2 cells. When the concentration of BAPTA-AM reached up to 30 μM and 40 μM, the consumption of glucose remarkably decreased by 54.79% and 78.60% compared to the cells only treated with CAP (Figure 4B). This indicates that the increase in calcium concentration mediates the promoting effect of capsaicin on the glucose metabolism of HepG2 cells.

### 2.4. Differential Gene Expression Profile of HepG2 Cells Induced by Capsaicin and Its Effect on Cell Glucose Utilization

To explore the molecular mechanisms of capsaicin promoting the glucose consumption of HepG2 cells, RNA-seq was used for transcriptomic analyses. Three different time points (4 h, 12 h and 24 h) were set for transcriptome sequencing, and each time point was set separately for the DMSO control group and CAP treatment group. The high-quality raw data are shown in Appendix A. A total of 78, 146 and 507 DEGs (FPKM ≥ 1, log2(fold change) ≥ 1) were identified between the CAP treatment group and DMSO control group for 4 h (68 up- and 10 down-regulated), 12 h (121 up- and 25 down-regulated) and 24 h (241 up- and 266 down-regulated) of treatment, respectively (Figure 5A). A Venn diagram of DEGs showed that 21 DEGs were identical among all the three treatment group times. Through comparing DEGs of 4 h- and 12 h-treated cells, we found that 26 genes were identical, while 26 genes were the same in the 4 h- and 24 h-treated cells. In total, 100 genes had the same change trend between the 12 h- and 24 h-treated cells (Figure 5B). In light of Gene Ontology (GO) terms, biological process analysis revealed that the DEGs from 4 h-, 12 h- and 24 h-treated cells were primarily enriched in the regulation of cell death, the cell cycle, and the response to an extracellular or endogenous stimulus (Figure 5C). Moreover, the KEGG pathway analysis of DEGs were involved in 16 canonical pathways (Figure 5D). Among these, metabolic pathways, PI3K-AKT signaling pathway, carbon metabolism and the biosynthesis of amino acids might be responsible for the acceleration of capsaicin-induced glucose uptake and consumption of HepG2 cells.

Both GO and KEGG analyses indicated that capsaicin affected several metabolic processes in HepG2 cells. To investigate the possible regulation network among these genes, interaction networks that comprise all the DEGs were established for the 4 h-, 12 h- and 24 h-capsaicin-treated cells, respectively (Figure 6). It could be found that the tendency of the gene changes in 4 h-, 12 h- and 24 h-treated cells were quite consistent. The number of DEGs in the 12 h-treated HepG2 cells was more than in the 4 h-treated cells, with a very few exceptions like gene DDIT3 and SESN2. To further analyze the network, these genes were categorized into several pathway clusters, including intracellular signaling, transcription/translation, amino acid metabolism and lipid metabolism. As shown in Figure 6, we found that the clusters of amino acid metabolism were obviously up-regulated, while the numbers of up-regulated and down-regulated genes in intracellular signaling and transcription/translation were about the same.

### 2.5. Q-PCR Analysis of the Differentially Expressed Genes

As an approach to cross-check the reliability of RNA-seq data, the conventional quantitative PCR was usually utilized to assess the expression levels of several selected differentially expressed genes. In the 4 h-capsaicin-treated cells, the q-PCR validation results of three genes IGFBP3, NFX1 and SLFN5 were quite consistent with the RNA-seq. Eight genes selected from the 24 h-treated cells were validated by q-PCR. Most of them were involved in glycometabolism (like PCK2, ACSS2 and ASNS) and amino acid metabolism (like CTH and PSAT1). These genes were reported to play important roles in cell metabolism, and all were found to be up-regulated in capsaicin-treated cells. All the data demonstrated that gene expression changes identified by RNA-seq were consistent with those analyzed by q-PCR (Figure 7).

In conclusion, capsaicin may interact with TRPV1 on the surface of HepG2 cells, which induces the inward flow of calcium ions. The Ca^2+^ signal further activates signal pathways, such as PI3K/AKT, MAPK/ERK, etc. This eventually leads to increases in glucose uptake, consumption and ATP production (Figure 8).

## 3. Discussion

Obesity represents a major health challenge because it substantially increases the risk of metabolic diseases such as type 2 diabetes, steatohepatitis, hypertension and cancer [21]. Overfeeding and irregular living habits are major contributors to this disorder. Hence, dietary habits are of special interest to prevent and counteract obesity and its associated metabolic complications. Capsaicin, a major pungent ingredient in red pepper, has been widely reported to possess anti-obesity activity [5,22,23]. Many studies have reported that capsaicin affects the lipid and energy metabolism of 3T3-L1 preadipocytes and Caco-2 cells to exert anti-obesity activity [7,20,23,24]. In this study, we evaluated the impact of capsaicin on glucose uptake and consumption in HepG2 cells and delineated the gene expression changes involved. Our data showed that capsaicin accelerated the uptake and consumption of glucose in HepG2 cells. The gene expressions involved in intracellular signaling, transcription/translation, glycometabolism (like PCK2, ACSS2 and ASNS) and amino acid metabolism (like CTH and PSAT1) were remarkably changed after capsaicin treatment. This implied that the effect on glucose uptake and consumption on hepatocytes is a critical mechanism underlining the anti-obesity of capsaicin. The disturbance of glucose homeostasis is closely related to various metabolic syndromes, especially obesity [25]. The maintenance of glucose metabolism homeostasis is important for the body to keep normal biological functions. Mounting evidence has indicated that the consumption of CAP has a positive relationship with the amelioration of abnormal glucose metabolism, which has been further shown to ameliorate obesity [26]. In a diet-induced obesity animal model, CAP has also been proven to ameliorate obesity through increasing satiety, enhancing lipid and glucose metabolism, modulating the composition of gut microbiota, etc. [8]. The liver plays a key role in maintaining metabolic homeostasis. Hereon, the impact of capsaicin on glucose uptake and consumption in HepG2 cells and the delineated gene expression changes involved were evaluated. The results of the CCK-8 assay showed that capsaicin had no toxic effect on HepG2 cells. As for the glucose uptake, glucose consumption and ATP production, capsaicin showed a great stimulative effect on HepG2 cells. Transient receptor potential vanilloid type 1 (TRPV1) is a non-selective and ligand-gated cation channel which can be activated by multiple chemical stimuli, especially by capsaicin (an agonist of TRPV1) [17]. The activation of TRPV1 leads to a rapid cationic ionic influx of Ca^2+^, which then initiates a series of intracellular signaling pathways [27]. Through regulating TRPV1-mediated intracellular Ca^2+^ homeostasis, capsaicin has been reported to exert a range of biological activities, such as antitumor activity, antagonizing diabetic nephropathy and long-lasting analgesia activity, etc. [28,29,30]. Using Fluo3-AM to indicate the intracellular Ca^2+^ showed that the 24 h treatment of capsaicin dose-dependently increased the Ca^2+^ concentration of HepG2 cells. BATPA-AM, an intracellular calcium-chelating agent, blocking Ca^2+^-induced activation, remarkably inhibited the capsaicin-induced increase in the glucose consumption of HepG2 cells. In C2C12 skeletal muscle cells, capsaicin also has been reported to promote glucose uptake through activating calcium signaling [31]. These confirmed that capsaicin stimulated the glucose consumption of HepG2 cells through increasing intracellular Ca^2+^ by activating the TRPV1 channel. The calcium signaling pathway it regulated should be further studied. 

Research was carried out on the health effects of food-derived component benefits from holistic approaches to understand the underlining mechanisms. The transcriptome analysis in this study delineated hepatocyte gene expression differences between capsaicin-treated cells and the control. Furthermore, the expression of several target genes of interest were confirmed using real-time PCR. Manchanda reported that TRPV-1 activation by capsaicin drove the regulation of a wider range of lipid signaling molecules and was time dependent [32]. In the transcriptome analysis, the number DEGs of 4 h, 12 h and 24 h were 78, 146 and 507, respectively. This indicated that capsaicin regulated the gene expression of hepatocytes in a time-dependent manner, which was consistent with the conclusion of Manchanda [32]. The top canonical pathways influenced by capsaicin, compared with that of the control, were that of the metabolic pathways, PI3K-AKT signaling pathway and MAPK signaling pathway, etc. These pathways have been extensively reported to be involved in glucose homeostasis [33,34]. The genes which were reported to accelerate glucose uptake/consumption and ATP production, including phosphoenolpyruvate carboxykinase (PCK2), angiopoetin-like 4 (ANGPTL-4), IL-6, ETS translocation variant 5 (ETV5), Acetyl-coenzyme A synthetase (ACSS2), were significantly alerted in our RNA-seq gene expression data. With the extension of CAP processing time, the magnitude of the gene expression changes also increased. PCK2 is the rate-limiting enzyme in the metabolic pathway that produces glucose from lactate and other precursors derived from the citric acid cycle [35]. Verification results of q-PCR showed that in the 24 h-treated HepG2 cells, the gene expression of PCK2 was remarkably increased compared with the control, suggesting that CAP has important influence on the glucose metabolism of hepatocytes. Angiopoietin-like protein 4 (ANGPTL4) plays an important role in glucose metabolism and energy homoeostasis [36]. It is mainly released from the liver to act as an effective stimulant of muscle metabolism in response to meeting the energy requirement of exercise [37]. In our RNA sequencing results, CAP treatment could dramatically promote the expression of ANGPTL4 in HepG2 cells (Figure 6). Further validation of this change could be conducted in further studies. IL-6 is an important contributor to glucose and energy homeostasis. The definite role of IL-6 (beneficial or detrimental) on glucose metabolism is still debated. Generally, IL-6 was viewed to be detrimental to glucose homeostasis based on epidemiological, in vitro and in vivo studies. However, growing evidence indicates that increased IL-6 levels might play a positive role in improving glycemic control [38,39]. After the 12 h and 24 h CAP treatment, the mRNA expression of IL-6 was remarkably increased in HepG2 cells from the results of RNA sequencing and q-PCR. This indicated that the increased IL-6 may play an important role in a CAP-mediated increase in glucose uptake and consumption. VEGFA (vascular endothelial growth factor (A) is a growth factor active in angiogenesis, vasculogenesis and endothelial cell growth [40]. Up-regulation of VEGFA was reported to be involved in a GLUT 1-mediated increase in glucose uptake and metabolism [41]. The expression of the VEGFA gene was up-regulated in three different time-treated cells, based on the RNA-seq data. Moreover, our q-PCR results verified this in 12 h- and 24 h-treated hepatocyte cells. These indicated that an increased expression of VEGFA participated in a CAP-mediated increase in glucose uptake and consumption. In addition to the genes associated with glucose metabolism, the expression of genes involved in amino acid metabolism also changed after CAP treatment. CTH (Cystathionine gamma-lyase), a major H2S-producing enzyme with L-cysteine, has been reported to participate in liver glucose generation and metabolism via post-translation modification [42]. From the RNA-seq assay, the gene expression of CTH in the 4 h-, 12 h- and 24 h-CAP-treated group was up-regulated. The verification of the mRNA expression of CTH using q-PCR was constant with a result of RNA-seq. These results suggest that not only genes related to glucose metabolism are involved in the CAP-induced acceleration of glucose uptake and consumption, but genes related to amino acid metabolism also participate in this process. 

In conclusion, the present study demonstrated that CAP remarkably promoted the glucose uptake and consumption of hepatocytes. The increase in intracellular Ca^2+^ was involved in this process. The DEG analysis allowed for us to establish a gene database to focus on biological processes and pathways that might be related to the CAP-induced glucose uptake and consumption. These results encourage further studies to be carried out on the effect of CAP on the glucose metabolism of hepatocytes and liver tissue.

## 4. Materials and Methods

### 4.1. Cell Culture

HepG-2 cells were obtained from China Center for Type Culture Collection (CTCC, Wuhan, China), cultured in Dulbecco’s modified Eagle’s medium (DMEM, Hycolne) and supplemented with 10% fetal calf serum (Si Jiqing, Zhejiang Tianhang Biotechnology Corporation, Jinhua, China) with 1% penicillin–streptomycin (Genview, Houston, TX, USA) at 37 °C in a humidified atmosphere of 5% CO_2_.

### 4.2. Cell Viability Assay

Cell viability was measured by CCK-8 assays according to the instructions previously reported. Briefly, HepG2 cells were seeded at a density of 1.0 × 10^4^ cells/well on 96-well cultivation plates. After attachment, the cells were treated with capsaicin (Sigma, Gillingham, UK) (25, 50, 100 and 200 µM dissolved in DMSO) for 24 h. The control was performed under the same condition but in the absence of capsaicin. Wells with no cells were selected as the blank control. The percentages of viable cells were determined by CCK-8 assays (Dojindo, Kumamoto, Japan), as described. The cell viability was calculated by the following formula:Cell viability (%) = [(mean experiment absorbance − mean blank control absorbance)**/**(mean control absorbance − mean blank control absorbance)] × 100

### 4.3. Glucose Consumption Assay

HepG2 cells were seeded at a density 1.0 × 104 cells/well. The cells were treated with capsaicin at various concentrations (25, 50, 100 and 200 µM dissolved in DMSO) for 24 h. When studying the time effect of capsaicin on the glucose consumption of HepG2 cells, the cells were treated with 100 µM of capsaicin for 4 h, 8 h, 12 h and 24 h. After the treatment, glucose content in the culture medium was measured by the glucose oxidase method using a glucose assay kit (Shanghai Mind Bioengineering Co., Ltd., Shanghai, China) following the manufacturer’s instructions. The OD values of samples were measured by a Thermo scientific microplate reader at 500 nm.

### 4.4. Glucose Uptake Assay

HepG2 cells treated with capsaicin at various concentrations (25, 50, 100 and 200 µM dissolved in DMSO) for 24 h were washed with PBS. 2-NBDG (0.2 mM) (Sigma) was added into the culture medium for 30 min at 37 °C. The 2-NBDG fluorescence intensity was measured using a microplate reader at an excitation wavelength of 480 nm and an emission wavelength of 540 nm.

### 4.5. Measurement of Intracellular ATP Level

Adenosine triphosphate (ATP) was assessed by the firefly bioluminescence of an ATP assay kit (Beyotime, Shanghai, China). HepG2 cells were seeded into a 6-well plate at a density of 1.0 × 105 cells/well and then cultured to reach 80% confluence. The cells were pretreated with capsaicin at various concentrations (25, 50, 100 and 200µM dissolved in DMSO) and for different times (4 h, 8 h, 12 h and 24 h). They were then collected and lyophilized with 200 μL of an ice-cold ATP extraction buffer for 5 min. The suspension was scraped and then transferred into a microcentrifuge tube. Cell debris were removed by centrifugation at 12,000× *g* and 4 °C for 5 min. The supernatant was collected for further determination of intracellular ATP content using an ATP assay kit (Beyotime Biotechnology, Shanghai, China). The luminescence of an 80 µL sample with 100 µL of ATP detection buffer was assayed in a liquid scintillation counter (1450, PerkinElmer, Billerica, MA, USA). The ATP level was calculated from the ATP standard curve.

### 4.6. Measurement of the Intracellular Ca^2+^ Level

Intracellular Ca^2+^ levels were measured using Fluo-3AM (Dojindo, Kumamoto, Japan). Cells were cultured in confocal dishes and maintained for 12 h. Then, the cells were treated with different concentrations of capsaicin (25, 50 and 100µM) for 24 h. The cells were washed twice with PBS and incubated with 4 μM Fluo-3AM at 37 °C for 1 h in the dark. The cells were then washed twice with PBS and incubated for an additional 30 min at 37 °C. Then, the fluorescence signal of Ca^2+^ was measured by a confocal microscope (Olympus FV1200, Tokyo, Japan).

### 4.7. RNA Extraction

Total RNA was extracted from HepG2 cells treated with 100 µM of capsaicin for 4 h, 12 h and 24 h using Trizol reagent (Ambion, Life Technologies, Waltham, MA, USA). The concentration and purity of the RNA samples were examined at 260/280 nm ratio by Nanodrop (Thermo Scientific, Waltham, MA, USA). RNA degradation and contamination were monitored on 1% agarose gel electrophoresis (AGE) by the Mini-Sub Cell GT System (Bio-rad, Hercules, CA, USA).

### 4.8. RNA-Seq Analysis

HepG2 cells treated with capsaicin were placed in frozen storage tubes and put into a liquid nitrogen container quickly. Sequencing was performed with an Illunima NextSeq 500 by Appreciate the beauty of Life (BLife, Wuhan, China). The clean reads were obtained from the raw reads after removing the low-quality reads containing an adapter and poly-N. Clean reads from RNA-seq were mapped to human reference genome (GRCh38, https://asia.ensembl.org/index.html, accessed on 15 August 2021) with BOWTIE (version 2.0.0, Agilent, Santa Clara, CA, USA). The genes with log2(fold change) ≥ 1, FPKM ≥ 1 and adjusted *p*-value < 0.5 were set as the criteria for screening the differentially expressed genes (DEGs). Gene Ontology (GO) enrichment and Kyoto Encyclopedia of Genes and Genomes (KEGG) pathway analyses of the DEGs were implemented with the ClusterProfile R package, in which the gene length bias was corrected. For the gene interaction network analysis, the differential proteins based on the DEGs were submitted to STRING 9.0 (the Search Tool for the Retrieval of Interacting Genes/Proteins) to quantify the physical and functional interaction of these genes. The genes and their interaction were then uploaded to Cytoscope (Version 2.8.3) for data visualization.

### 4.9. Quantitative PCR Analysis

After HepG2 cells were treated with 100 μM of capsaicin for 4 h, 12 h and 24 h, total RNA was extracted using the Trizol reagent (Invitrogen, Carlsbad, CA, USA), according to the instructions provided. The cDNAs were synthesized from 2 µg of total RNA using M-MLV (Moloney murine leukaemia virus) reverse transcriptase (Promega, Shanghai, China). Real-time quantitative PCR analysis was performed using SYBR Green Realtime PCR Master Mix (Toyobo, Osaka, Japan) in a 10 μL reaction mixture. The thermal cycle conditions were as follows: 95 °C for 5 min, followed by 45 cycles of 95 °C for 15 s, and 60 °C for 30 s. Data were normalized to the results of actin and analyzed using the standard 2^−∆∆CT^ method. The specific primers of the genes used for quantitative PCR were listed in Appendix A.

### 4.10. Statistical Analysis

The results were represented as mean ± SD (Standard Deviation). We checked the data for Gaussian distribution by the Shapiro–Wilk test. The statistical significance of the data with Gaussian distribution were determined by the one-way analysis of variance (ANOVA) and *t*-test, while the other data were analyzed using a non-parametric test. Statistical significance was defined as *p* < 0.05 compared with the control values.

## 5. Conclusions

In conclusion, our results suggest that the capsaicin intervention efficiently accelerates the glucose uptake and consumption of hepatocytes (HepG2) in a dose- and time-dependent manner. The elevation of intracellular-free Ca^2+^ plays an important role in this process. The transcriptome analysis indicated that the activation of some canonical signaling pathways, like MAPK and PI3K-AKT, were thought to be the underling mechanism. The expression of genes involved in glucose and amino acid metabolism were also differently expressed between the capsaicin-treated cells and the control. As a potential food ingredient to alleviate obesity, deeper molecular mechanisms need to be further investigated.

## Figures and Tables

**Figure 1 molecules-28-05258-f001:**
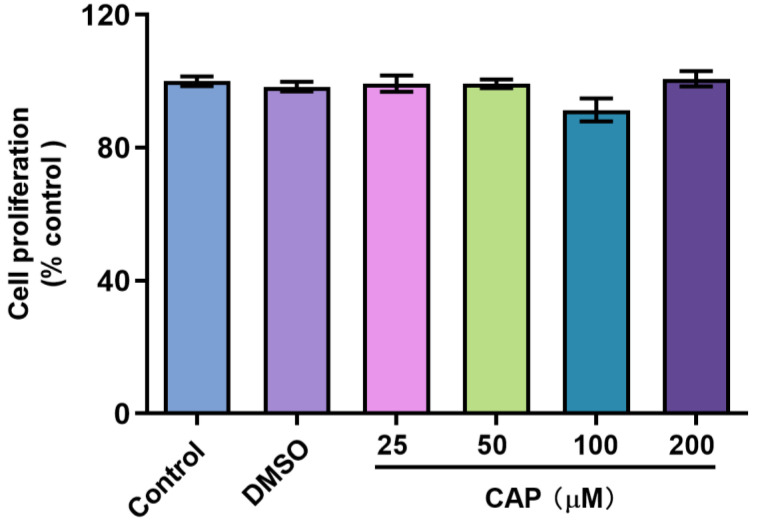
Effect of capsaicin on the proliferation of HepG2 cells. HepG2 cells were treated with capsaicin (25 μM, 50 μM, 100 μM, 200 μM in DMSO) for 24 h. The proliferation ratios of HepG2 cells were analyzed by the CCK-8 assay. The control group was treated with DMSO or DMEM alone. The data shown are the mean ± SD (Standard Deviation) of three independent experiments.

**Figure 2 molecules-28-05258-f002:**
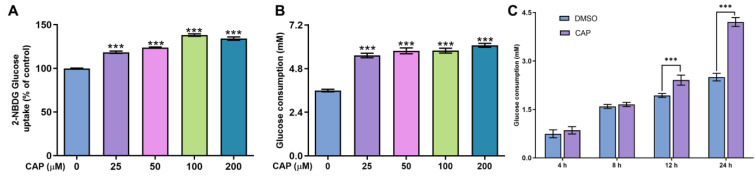
Effects of capsaicin on the glucose concentration and uptake of HepG2 cells. (**A**) Effects of capsaicin (25, 50, 100 and 200 μM) on the 2-NBGD glucose uptake level of HepG2 cells. HepG2 cells were incubated with different concentrations of capsaicin for 24 h to detect the basal glucose consumption. DMSO was used as a negative control. (**B**) The effect of capsaicin on the basal glucose consumption of HepG2 cells. (**C**) HepG2 cells were treated with 100 μM of capsaicin for different lengths of time (4, 8, 12 and 24 h). The basal glucose consumption was detected. The data shown are the mean ± SD (Standard Deviation) (*** *p* < 0.001) of three independent experiments.

**Figure 3 molecules-28-05258-f003:**
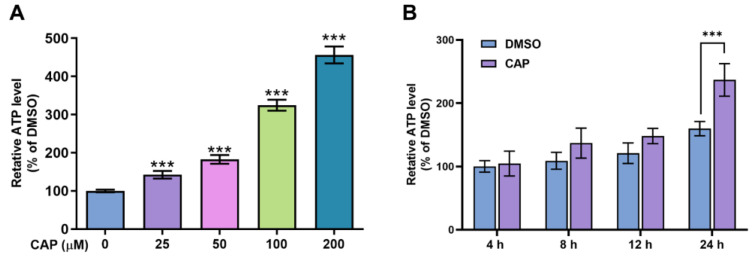
The effects of capsaicin on intracellular ATP levels. (**A**) HepG2 cells were treated with a range of capsaicin concentrations (25, 50, 100 and 200 μM) for 24 h. (**B**) HepG2 cells were treated with 100 μM of capsaicin for 24 h. Intracellular ATP was measured by an ATP assay kit. The data shown are the mean ± SD (Standard Deviation) (*** *p* < 0.001) of three independent experiments.

**Figure 4 molecules-28-05258-f004:**
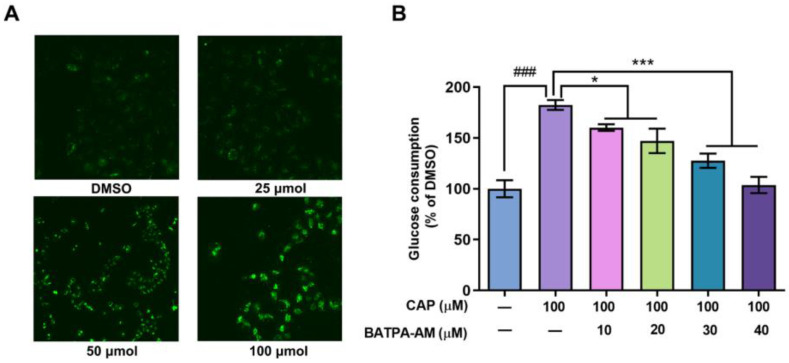
Ca^2+^ was involved in the capsaicin-induced glucose metabolism increase in HepG2 cells. (**A**) HepG2 cells were treated with different concentrations of capsaicin (25, 50 and 100 μM) for 24 h. The intracellular Ca^2+^ was detected by measuring the fluorescence intensity of the calcium indicator Fluo-3AM with fluorescence microscopy. (**B**) The glucose consumptions of HepG2 cells after treatment with capsaicin alone or capsaicin and BATPA-AM (a calcium-chelating agent) were measured using a glucose assay kit (* *p* < 0.05, *** *p* < 0.001 vs. CAP 100 μM; ### *p* < 0.001 vs. control).

**Figure 5 molecules-28-05258-f005:**
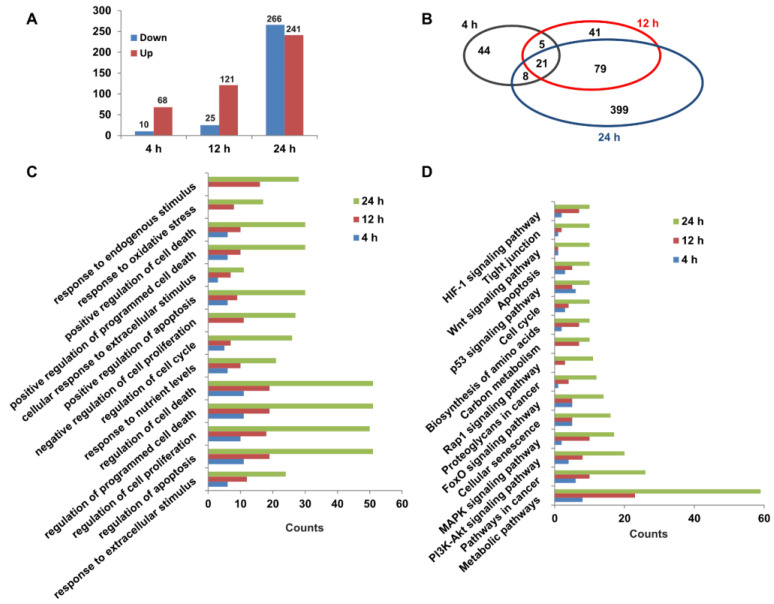
Transcriptomic changes in HepG2 cells exposed to capsaicin. (**A**) Summary of up-/down-regulated DEGs numbers between capsaicin-treated groups and the control groups at different times (4 h, 12 h and 24 h). (**B**) The Venn diagram shows an overlap of DEGs between different capsaicin-treated times (4 h, 12 h and 24 h). (**C**) The DEGs of HepG2 cells at different capsaicin-treated times (4 h, 12 h and 24 h) were enriched in 14 GO terms. (**D**) The DEGs of HepG2 cells at different capsaicin-treated times (4 h, 12 h and 24 h) were mapped in KEGG pathways. Pathways including at least 3 mapped proteins were considered.

**Figure 6 molecules-28-05258-f006:**
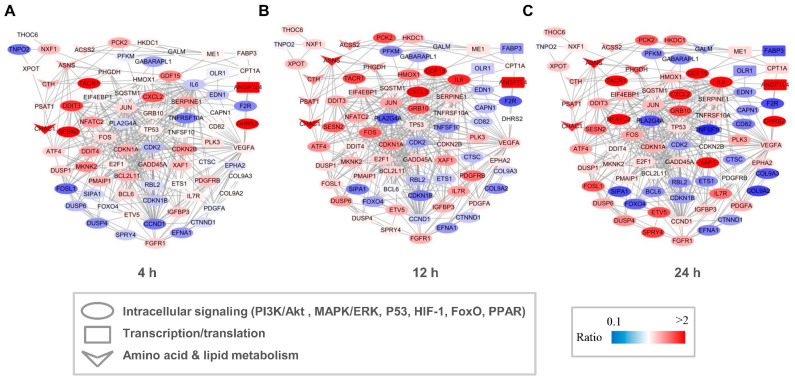
Network of DEGs of HepG2 cells after treatment with capsaicin. Multiple signal pathways including intracellular signaling, transcription, translation, amino acid metabolism and lipid metabolism were involved in the DEG interaction network after 4 h (**A**), 12 h (**B**) and 24 h (**C**) capsaicin treatment. Node color represents the quantitative ratio.

**Figure 7 molecules-28-05258-f007:**
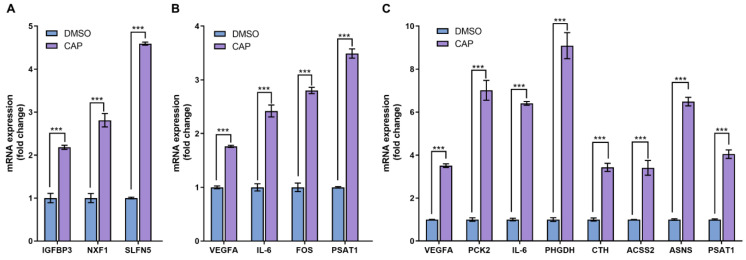
Real-time PCR analysis of differentially expressed genes. (**A**) At the 4 h-capsaicin-treated group, the mRNA expression of IGFBP3, NXF1 and SLFN5 were selected to be validated by real-time PCR. (**B**) At the 12 h-capsaicin-treated group, the mRNA expression of VEGFA, IL-6, FOS and PSAT1 were selected to analyzed by real-time PCR. (**C**) At the 24 h-capsaicin-treated group, the mRNA expression of VEGFA, IL-6, PCK2, PHGDH, CTH, ACSS2, ASNS and PSAT1 were selected to be analyzed by real-time PCR (*** *p* < 0.001 vs. DMSO).

**Figure 8 molecules-28-05258-f008:**
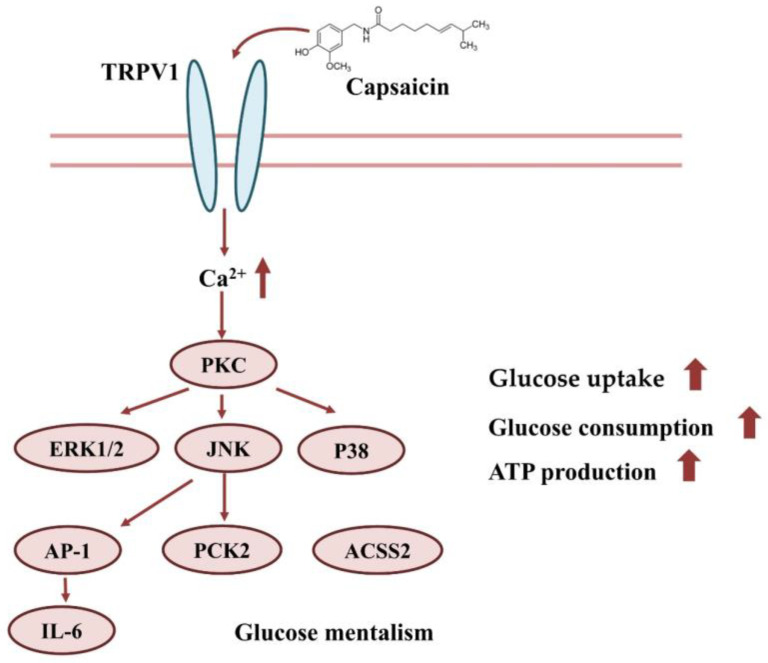
Effect of capsaicin on glucose uptake and the consumption of HepG2 cells.

## Data Availability

The data presented in this study are available on request from the corresponding author.

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
