# Peer review of "Effects of Capsaicin on Glucose Uptake and Consumption in Hepatocytes"

_molecules, 2023, doi:10.3390/molecules28135258_

Round 1
Reviewer 1 Report
In the present study Capsaicin, the key ingredient in Capsicum species, was shown to increases glucose uptake and ATP production in liver cells. This effect may be linked to a rise in intracellular calcium uptake. Genetic analysis shows alterations in several genes related to glucose and amino acid metabolism. Thus, they claimed capsaicin could potentially enhance hepatocyte glucose management. Investigations are very interesting and the study comprehensive but, there are several concerns related to the manuscript.
1- Some introductory information and theory of the study should be presented in the abstract part.
2- Introduction part is seemed very short. There should be more information related to hepatocyte glucose mechanism and capsaicin effects. One more paragraph can be added.
3- Lack of in vivo investigations of capsaicin in glucose metabolism and relationship with obesity should be mentioned in the discussion part and possible animal model for this type of study should be discussed.
Overall, the study is well designed and results were confirmed especially RNAseq verified with qPCR. After a revision the manuscript can be accepted.
Author Response
Response to Reviewer 1 Comments
In the present study Capsaicin, the key ingredient in Capsicum species, was shown to increases glucose uptake and ATP production in liver cells. This effect may be linked to a rise in intracellular calcium uptake. Genetic analysis shows alterations in several genes related to glucose and amino acid metabolism. Thus, they claimed capsaicin could potentially enhance hepatocyte glucose management. Investigations are very interesting and the study comprehensive but, there are several concerns related to the manuscript.
Response:Thank you for taking the time to process the submission of our original paper entitled “Effects of capsaicin on glucose uptake and consumption in hepatocyte”. We believe that your detailed and constructive suggestions help us to make the revised manuscript much better that the 1 st round. We have made efforts to address all the concerns raised.
1- Some introductory information and theory of the study should be presented in the abstract part.
Response:Thank you for this kind suggestion. Some introductory information and theory of the study has been added in the abstract of the revised manuscript.
“Obesity represents a major health challenge for it substantially increasing the risk of metabolic disease. Capsaicin, the major active ingredient of Capsicum spp., has been reported to possess anti-obesity activity.” (Page 1, lines 21-23)
2- Introduction part is seemed very short. There should be more information related to hepatocyte glucose mechanism and capsaicin effects. One more paragraph can be added.
Response:We appreciate this kind suggestion. In the revised manuscript, one more paragraph, which was related to hepatocyte glucose mechanism and capsaicin effects, has been added to make the introduction more informative.
“The transient receptor potential channel vanilloid type 1 (TRPV1), which was expressed in many tissues, is a member of the transient receptor potential (TRP ) channel superfamily. The activation of TRP have been shown to be involved in a wide range of metabolic process, including glucose metabolism and progression of obesity [13]. The liver plays an essential role in nutrient metabolism for that it has first access to the ingested nutrients by virtue of their absorption into the hepatic portal vein. It is the main metabolic organ, especially in glycometabolism. The dysregulation of hepatic glucose metabolic pathways is central to many pathologies, including obesity and even type 2 diabetes [14]. It has demonstrated that TRPV1 is a mechanosensitive ion channel that has been shown to regulate the energy expenditure and cholesterol metabolism in hepatocytes [15,16]. As a highly selective agonist for TRPV1, CAP exerts many of its biological activities through TRPV1 activation [17]. It was also reported to exerts action on TRPV1 channel to mediate the regulation of glucose homeostasis and obesity [18].” (Page 2, lines 58-71)
3- Lack of in vivo investigations of capsaicin in glucose metabolism and relationship with obesity should be mentioned in the discussion part and possible animal model for this type of study should be discussed.
Response: Thank you for your suggestion. In the revised manuscript, the in vivo investigation of capsaicin in glucose metabolism and relationship with obesity has been added in the discussion part. The possible animal model for this type of study was also discussed in the revised manuscript.
“Disturbance of glucose homeostasis is closely related to various metabolic syndromes, especially obesity [25]. Maintenance of glucose metabolism homeostasis is important for the body to keep normal biological functions. Mounting evidence has indicated that the consumption of CAP has positive relation to the amelioration of abnormal glucose metabolism, which has been further shown to ameliorate obesity [26]. In diet induced obesity animal model, CAP has also been proofed to ameliorate obesity through increasing satiety, enhancing the lipid and glucose metabolism, modulating the composition of gut microbiota and etc [26]. The liver plays a key role in maintaining metabolic homeostasis. Hereon, the impact of capsaicin on glucose uptake and consumption in HepG2 cells and delineated the gene expression changes involved were evaluated.” (Page 9-10, lines 259-268)
Overall, the study is well designed and results were confirmed especially RNAseq verified with qPCR. After a revision the manuscript can be accepted.
Response: Thank you again for your recognition and constructive suggestions for our study. We have thoroughly revised the manuscript according to your suggestions.

Reviewer 2 Report
The paper entitled ‘Effects of capsaicin on glucose uptake and consumption in 2 hepatocyte’ concerns interesting topic of capsaicin action on glucose consumption in hepatocytes, which seems important from the point of view of obesity, as well as diabetes in humans. There are reports showing that capsaicin, or wider, chili peppers consumption has positive effect on health, preventing among others from development of metabolic syndrome. The paper is well written, transparent, with precisely described scheme of research in a logic manner.
Below are some minor suggestions
Line 42
‘with relatively low toxicity’
Capsaicin can have positive impact on health, but there are reports showing its potential toxicity.
For example:
1/ high dose chili extracts impaired gut permeability and induced gut dysbiosis (Panpetch W, Visitchanakun P, Saisorn W, Sawatpanich A, Chatthanathon P, Somboonna N, et al. (2021) Lactobacillus rhamnosus attenuates Thai chili extracts induced gut inflammation and dysbiosis despite capsaicin bactericidal effect against the probiotics, a possible toxicity of high dose capsaicin. PLoS ONE 16(12): e0261189. https://doi.org/10.1371/journal.pone.0261189)
2/ increased incidence of gastric cancer amongst high chili pepper consumers in Mexico who were estimated to consume 90–250 mg of capsaicin per day; others have reported gall bladder cancer patients in Chile consume significantly higher amounts of red chili pepper relative to controls (Bley, K., Boorman, G., Mohammad, B., McKenzie, D., & Babbar, S. (2012). A comprehensive review of the carcinogenic and anticarcinogenic potential of capsaicin. Toxicologic pathology, 40(6), 847-873.)
The authors should indicate in the Introduction the possible toxic effects of capsaicin on health.
Line 160: space between 241_up
Line 223: shouldn’t it be affected rather than effected?
Line 258: two dots after etc
Line 290: mentalism – should be changed to metabolism
Line 388: Statistical analysis
Whether the data meets the assumptions of ANOVA? How was the normality of data calculated?
Author Response
Response to Reviewer 2 Comments
The paper entitled ‘Effects of capsaicin on glucose uptake and consumption in hepatocyte’ concerns interesting topic of capsaicin action on glucose consumption in hepatocytes, which seems important from the point of view of obesity, as well as diabetes in humans. There are reports showing that capsaicin, or wider, chili peppers consumption has positive effect on health, preventing among others from development of metabolic syndrome. The paper is well written, transparent, with precisely described scheme of research in a logic manner.
Response:Thank you for taking the time to process the submission of our original paper entitled “Effects of capsaicin on glucose uptake and consumption in hepatocyte”. We believe that your detailed and constructive suggestions help us to make the revised manuscript much better that the 1 st round. We have made efforts to address all the concerns raised.
Below are some minor suggestions
- Line 42
‘with relatively low toxicity’
Capsaicin can have positive impact on health, but there are reports showing its potential toxicity.
For example:
1/ high dose chili extracts impaired gut permeability and induced gut dysbiosis (Panpetch W, Visitchanakun P, Saisorn W, Sawatpanich A, Chatthanathon P, Somboonna N, et al. (2021) Lactobacillus rhamnosus attenuates Thai chili extracts induced gut inflammation and dysbiosis despite capsaicin bactericidal effect against the probiotics, a possible toxicity of high dose capsaicin. PLoS ONE 16(12): e0261189. https://doi.org/10.1371/journal.pone.0261189)IF: 3.752 Q2
2/ increased incidence of gastric cancer amongst high chili pepper consumers in Mexico who were estimated to consume 90–250 mg of capsaicin per day; others have reported gall bladder cancer patients in Chile consume significantly higher amounts of red chili pepper relative to controls (Bley, K., Boorman, G., Mohammad, B., McKenzie, D., & Babbar, S. (2012). A comprehensive review of the carcinogenic and anticarcinogenic potential of capsaicin. Toxicologic pathology, 40(6), 847-873.)
The authors should indicate in the Introduction the possible toxic effects of capsaicin on health.
Response: Thank you for your kind suggestion. The information related to the possible toxic effects of capsaicin on health has been added in the revised manuscript.
“Definitely, inappropriate dosage of CAP ingestion has adverse health effects in human subjects, like gastric ulcers, hyperalgesia, gut inflammation and even promoting the development of tumors [9-12].” (Page 2, Lines 55-57)
Reference:
[9] Bode, A. M.; Dong, Z., The Two Faces of Capsaicin. Cancer Research 2011, 71, (8), 2809-2814.
[10] Panpetch, W.; Visitchanakun, P.; Saisorn, W.; Sawatpanich, A.; Chatthanathon, P.; Somboonna, N.; Tumwasorn, S.; Leelahavanichkul, A., Lactobacillus rham- nosus attenuates Thai chili extracts induced gut inflammation and dysbiosis despite capsaicin bactericidal effect against the probiotics, a possible toxicity of high dose capsaicin. PLoS One, 2021, 16, (12), e0261189.
[11] Bley, K.; Boorman, G.; Mohammad, B.; McKenzie, D.; Babbar, S., A compre- hensive review of the carcinogenic and anticarcinogenic potential of capsaicin. Toxicol Pathol, 2012, 40, (6), 847-73.
[12] Lu, M.; Chen, C.; Lan, Y.; Xiao, J.; Li, R.; Huang, J.; Huang, Q.; Cao, Y.; Ho, C. T., Capsaicin-the major bioactive ingredient of chili peppers: bio-efficacy and delivery systems. Food Funct, 2020, 11, (4), 2848-2860.
- Line 160: space between 241_up
Response: We are sorry for our careless mistake. In the revised manuscript. The space between 241 and up has been added. “241 up” (Page 6, Line 180).
- Line 223: shouldn’t it be affected rather than effected?
Response: We are sorry for our careless mistake. The “effected” has been revised to “affected” (Page 9, Line 250).
- Line 258: two dots after etc
Response: We are sorry for our careless mistake. It has been revised to “etc.” (Page 10, Line 297).
- Line 290: mentalism – should be changed to metabolism
Response: We are sorry for this careless mistake. It has been revised to “metabolism” (Page 11, Line 329).
- Line 388: Statistical analysis
Whether the data meets the assumptions of ANOVA? How was the normality of data calculated?
Response: Thank you for your question. We have checked the data for Gaussian distribution by Shapir-Wilk test before statistical analysis. The date with P values > 0.5 are Gaussian distribution. In the data analysis, only the Gaussian distributed data were analyzed using one-way ANOVA. We are sorry about that we did not clearly describe the data analysis. In the revised manuscript, we have added the following information to make it clear.
“We checked the data for Gaussian distribution by Shapir-Wilk test. The statistical significance of the data with Gaussian distribution were determined by one-way analysis of variance (ANOVA) and t-test. While the other data were analyzed using non-parametric test.” (Page13, Lines 428-431)

Reviewer 3 Report
Article Review
The study “Effects of capsaicin on glucose uptake and consumption in hepatocytes examines the impact of capsaicin on glucose metabolism in hepatocytes, exploring its association with ATP production and elevated calcium ion levels. Through an analysis of the manuscript, I have identified several areas for improvement that would enhance the scientific rigor and clarity of the article.
Comments:
Formatting and spelling:
Should the Latin name of pepper be in italics? As well as “in vivo” & “in vitro”?
In Figures 2 and 4, the vertical axis is labeled as "clucose" instead of "glucose." Please correct this error.
What does the abbreviation "SEM" stand for in the figure captions? Shouldn't it be "SD"? Please expand this abbreviation throughout the text.
Precision and clarity:
Is glucose absorbed only by hepatocytes? Please explain why the authors chose this specific cell model and provide additional information in the introduction.
In the results section, I noticed that the titles are formulated as statements, e.g., "Capsaicin affects glucose metabolism." Would it be more appropriate to use a neutral formulation, such as "The impact of capsaicin on glucose metabolism"? Please consider improving these titles.
The results section contains mental shortcut, such as line 70. Please provide a clear relationship between glucose uptake and fluorescence intensity, avoiding the interchangeable use of these terms. It is also necessary to explain the test earlier.
The authors use various abbreviations, such as "KEGG DEG pathway," "GO," and others, without prior explanation. Please provide explanations for these abbreviations throughout the text to ensure reader comprehension.
Additional information:
I encourage the authors to include a figure or diagram illustrating the impact of capsaicin on glucose uptake by liver cells. This will facilitate readers' understanding of the presented mechanism.
Please provide a concise description of the bioinformatics tools used in the analysis.
In-depth discussion and literature:
The article lacks an in-depth discussion with the existing literature or information on the absence of such literature. In the context of the presented results and conclusions, it would be valuable to reference relevant scientific publications that confirm or support the obtained results. Please consider supplementing the article with references to appropriate research works.
Summary:
In conclusion, the provided comments aim to improve the quality and scientific value of the article. Please consider incorporating these suggestions before submitting the article for publication. Additionally, I encourage you to explore the literature relevant to the research topic and expand the discussion in the context of existing scientific studies.
-
Round 2
Reviewer 1 Report
The manuscript can be accebted in the present form.
Reviewer 3 Report
The authors of the manuscript have addressed all of my previous comments. The expanded discussion and the inclusion of a diagram depicting the impact of capsaicin on glucose metabolism have significantly enhanced the clarity of the presented content. I appreciate the detailed responses provided to all of the comments.
I have no further remarks regarding the content.